# Incidence of Cholangitis and Sepsis Associated with Percutaneous Transhepatic Cholangiography in Pediatric Liver Transplant Recipients

**DOI:** 10.3390/antibiotics10030282

**Published:** 2021-03-10

**Authors:** Naire Sansotta, Ester De Luca, Emanuele Nicastro, Alessandra Tebaldi, Alberto Ferrari, Lorenzo D’Antiga

**Affiliations:** 1Paediatric Hepatology, Gastroenterology and Transplantation, Hospital Papa Giovanni XXIII, 24127 Bergamo, Italy; enicastro@asst-pg23.it (E.N.); ldantiga@asst-pg23.it (L.D.); 2Department of Pediatrics, University of Milano Bicocca, 20126 Milan, Italy; e.deluca@campus.unimib.it; 3Infectious Diseases Unit, Hospital Papa Giovanni XXIII, 24127 Bergamo, Italy; atebaldi@asst-pg23.it; 4FROM Research Foundation, Statistics, Hospital Papa Giovanni XXIII, 24127 Bergamo, Italy; aferrari34@yahoo.com

**Keywords:** children, liver transplant, cholangiography

## Abstract

**Background.** Percutaneous transhepatic cholangiography (PTC) is an established treatment in the management of biliary strictures. The aim of our study was to determine the incidence of PTC-related infectious complications in transplanted children, and identify their precise aetiol-ogy. **Methods.** We retrospectively reviewed all PTC performed from January 2017 to October 2020 in our center. Before the procedure, all patients received antibiotic prophylaxis defined as first line, while second line was used in case of previously microbiological isolation. Cholangitis was defined as fever (>38.5°) and elevated inflammatory markers after PTC, while sepsis included hemodynamic instability in addition to cholangitis. **Results.** One hundred and fifty-seven PTCs from 50 pediatric recipients were included. The overall incidence of cholangitis and sepsis after PTC was 44.6% (70/157) and 3.2% (5/157), respectively, with no fatal events. Blood cultures yielded positive results in 15/70 cases (21.4%). *Enterococcus faecium and Pseudomonas aeruginosa* were the most common isolated pathogens. Multidrug-resistant (MDR) pathogens were found in 11/50 patients (22%). **Conclusion.** PTC is associated with a relatively high rate of post-procedural cholangitis, although with low rate of sepsis and no fatal events. Blood cultures allowed to find a precise aetiology in roughly a quarter of the cases, showing prevalence of *Enterococcus faecium and Pseudomonas aeruginosa*.

## 1. Introduction

Biliary complications are a major source of morbidity after liver transplant (LT), and their reported incidence varies between 5 and 20% in children [1,2]. If properly managed, biliary complications are thought not to affect significantly patient and graft survival [3].

Percutaneous transhepatic cholangiography (PTC) has taken a decisive role in diagnosing biliary stricture in pediatric LT recipients, as it is considered the gold standard for identifying and quantifying the stenosis [4]. The percutaneous access allows also to treat these patients through balloon dilatation. PTC has shown low incidence of major complications, with high success rates in adults. Cholangitis and sepsis represented minor issues, easily treated in most patients [5]. In adults the risk of sepsis following biliary interventional procedures ranges from 0.8 to 2.3%. An international consensus panel defined an upper limit of a five percent sepsis rate as a quality parameter for percutaneous transhepatic biliary drainage (PTBD) [6].

Bilioenteric anastomosis, previous biliary procedures, and obstructive jaundice are associated with an increased risk of positive bile culture or sepsis. Once a drainage is placed through a bilioenteric anastomosis, the distal portion of the catheter is in the small bowel, thus favoring the migration of enteric organisms into the biliary tree. *Enterococcus* spp., *Candida* spp., gram negative aerobic bacilli, *Streptococcus viridans*, *E. coli*, *Clostridium*, *Klebsiella* spp., *Pseudomonas* spp., *Enterobacter cloacae* are the organisms often isolated in an infected biliary tree [7].

Given the high rate of infectious complications, which can be life-threatening, the Society of Interventional Radiology (SIR) clinical Practice Guidelines recommended a prophylactic antibiotic administration before PTC [8]. According to the guidelines, in the absence of prior positive culture results from bile to guide antimicrobial therapy, common empiric IV antibiotic choices (first line antibiotic prophylaxis) are preferred. These include ceftriaxone, ampicillin/sulbactam, ampicillin, and gentamycin (or if penicillin-allergic, vancomycin or clindamycin with an aminoglycoside). Furthermore, in agreement with Tokyo guidelines [9], prophylactic therapy should be selected depending on local antimicrobial susceptibility. In particular, the choice should carefully take into account the incidence of extended-spectrum beta-lactamases (ESBL) and carbapenemase-producing bacteria [10].

The incidence of cholangitis and sepsis associated with PTC—especially in transplanted children—is not well characterized. The aim of our study was to determine the incidence of PTC-related infectious complications and identify the microbiological aetiology in transplanted children.

## 2. Results

### 2.1. Clinical Features of the Study Population

A total of 157 PTC (99 de novo and 58 exchange) from 50 pediatric recipients were performed in our center in the study period. Demographic and clinical features of study population are described in Table 1.

Female to male ratio was 1:1 and the median age at the time of PTC was 3.05 years (IQR:3–9).

The indications for liver transplant (LT) were biliary atresia (65%), genetic cholestasis (14%), metabolic disease (2%), and miscellaneous (19%). Biliary reconstruction was most frequently performed with a Roux-en-Y hepaticojejunostomy (96%), while a duct-to-duct anastomosis was performed in the remaining 4% of our patients.

In our study population, out of 50 transplanted children, 21 (42%) patients in 57 PTC (36%) presented with at least one microbiological isolation (blood or bile culture) in their past medical history before the procedure. Overall, 11 patients (22%) were found to have microbiological isolation for multidrug resistant pathogens (MDR).

### 2.2. Antibiotic Prophylaxis and Infectious Related Complications

First-line antibiotic prophylaxis was administered in 135 procedures (86%), while second line prophylaxis was needed in 22 cases (14%). Of note, among first line prophylaxis, Cefotaxime was the most common used antibiotic in 73 procedures (54%), followed by Piperacillin-Tazobactam in 46 cases (34%), whereas Ciprofloxacin was administrated in 16 PTC.

The overall incidence of cholangitis and sepsis after PTC was 44.6 and 3.2%, respectively.

The incidence of cholangitis was 52.7% (39/74) in de novo group and 37.3 % (31/83) in the exchange group, with a clear trend to statistical significance (*p* = 0.053).

The incidence of sepsis was 2.7% (2/74) in de novo group and 3.6 % (3/83) in the exchange group (*p* = 0.745). Bilioplasty was found to be associated with higher risk of cholangitis and sepsis (*p* = 0.003), while patient age, concomitant use of steroids and the presence of central line were not significantly associated with infectious complications, *p* = 0.21, *p* = 0.06, *p* = 0.86, respectively. Interestingly, we did not find any significant correlation between type of antibiotic used as first line prophylaxis and risk of cholangitis and sepsis (*p* = 0.166). Furthermore, the previous microbiological isolation was not a risk factor for the following procedures, see Table 2.

### 2.3. Microbial Growth

Blood culture was performed in all PTCs complicated by cholangitis and sepsis (70/157), and it yielded positive results in 15/70 cases (21.4%). The isolated blood bacteria were gram-positive in 5/15 of cases (33%), while gram-negative were found in 12/15 of cases (80%). Two bacteria (Pseudomonas aeruginosa and Enterococcus faecium) were isolated in the same blood culture in 1 patient (6%).

Out of 70 infectious complications, bile culture was performed in 30 cases (42.3%), and at least one pathogen was isolated in 20/30 cases (66.6%).

Bile cultures revealed gram positive bacteria in 12/20 (60%) of cases, gram-negative in 11/20 (55%) of cases, while Candida albicans was isolated in 4/20 (20%). In half of our cases (10/20), two bacteria were isolated in the same bile sample.

Both bile and blood culture were positive in 5% of our procedures.

Overall, *Enterococcus faecium*, *Pseudomonas aeruginosa,* followed by *Escherichia coli* and *Klebsiella pneumoniae* were the most common isolated pathogens in blood and bile cultures (see Table 3).

Considering multidrug-resistant (MDR) pathogens, vancomycin-resistant *Enterococcus faecium* (VRE) were isolated in 3/50 (6%) transplanted children and carbapenemase producing *E. coli* in 1/50 patient (2%) in rectal swab, while ESBL germs (*K. pneumoniae, E. cloacae, E. aerogens, E. coli*) were found in 7/50 (14%) pediatric recipients either in blood or bile culture.

### 2.4. Antibiotic Therapy

Antibiotic therapy was continued in 115/157 procedures (73.2%) for the following reasons: cholangitis (70/115, 61%), sepsis (5/115, 4 %), previous colonization (22/115, 19 %), isolated elevated inflammation markers (10/115, 9%), or empirically (8/115, 7%).

First-line antibiotic treatment was administered after 81/115 procedures, while a second-line antibiotic therapy post procedural was required in 34/115 cases.

Remarkably, optimization antibiotic treatment was done in 22/115 (19%) cases, due to positive microbiological tests in 14/22 (64%) and persistent fever in 8/22 cases (36%).

Among second-line antibiotic treatment, alone or in association with first line antibiotics, the drugs used were Daptomycin and/or Linezolid in 6/115 cases (5.2%), Vancomycin in 13/115 cases (11.3%), Aminoglycosides (Gentamycin and Amikacin) in 4/115, Meropenem in 10/115. Antifungal therapy included Fluconazole and Micafungin, and it was administered in 2/115 cases.

Remarkably, Carbapenemase producing *E.coli* was treated with a combination of high dose Meropenem and Tigecycline, while in case of *Vancomycin-resistant Enterococcus spp*. *(VRE)* Daptomycin or Linezolid were needed.

Second line antibiotic treatment post procedure was more common in case of de novo PTC compared with an exchange procedure (*p* = 0.01). The mean duration of antibiotic therapy following post-procedural complications was 9.6 days. None of these patients required admission to intensive care unit. Antibiotic strain susceptibility was described in Appendix A.

## 3. Discussion

Percutaneous transhepatic cholangiography (PTC) was shown to promote infections of the biliary tree [11], with a risk of cholangitis and sepsis exceeding 20%. However, in adults, this risk was reduced to less than 2% by the use of prophylactic antibiotics [12].

In adults with neoplastic or benign biliary stenosis, the incidence of cholangitis and sepsis despite antibiotic prophylaxis before the endoscopic procedures such as PTC or ERCP (endoscopic-retrograde-cholangio-pancreatography) are extremely variable, with a reported rate ranging from 1 to 38% and from 0.8 to 2.3%, respectively [13,14]. The incidence of these complications in children are unknown and microbiology data are very scarce [11].

To the best of our knowledge, this is the first pediatric study evaluating the incidence of PTC-related infectious complications and identifying the microbiological culprit in transplanted children.

According to the current guidelines [10] available for adults, we started antimicrobial therapy before all PTC procedures. First line antibiotic prophylaxis was adopted in the majority of our procedures (86%), while second-line prophylaxis was started in selected cases based on previously isolated pathogens. The overall incidence of cholangitis and sepsis after PTC was 44.6 and 3.2%, respectively. As compared with adult studies [5], our children showed an increased risk of cholangitis. That finding was likely due to the presence of bilio-enteric anastomosis (Roux-en-Y hepaticojejunostomy) done in 96% of our study population, which may play a role in ascending cholangitis.

Blood culture yielded positive results in 15/70 (21.4%) infectious PTC-related complications, in agreement with previous studies that found positive rates of blood cultures ranged from 21 to 71% in adults with acute cholangitis [15].

Bile culture grew at least one pathogen in 20/30 cases (66.6%), while both bile and blood culture were positive in 5% of our procedures. Rosch et al. found bacteria in 60% of cases during the initial biliary drainage placement, and 24 h later this rate had already increased to 85%; two or more microorganisms were found initially in 40% and after a few days in 70%. During later PTC exchanges, bacteribilia increased to 100%, but the rate of clinical sequelae decreased dramatically [16]. Unfortunately, high study heterogeneity and small sample size made these results inconclusive. Furthermore, cultures from drainage fluid might secondarily yield microorganisms that colonize the drainage tubing and do not necessarily play a pathogenic role [11].

Overall, *Enterococcus faecium*, *Pseudomonas aeruginosa,* followed by *Escherichia coli* and *Klebsiella pneumoniae* were the most common isolated pathogens in blood and bile cultures, as shown in other studies [16,17]. As expected, *Staphylococcus aureus* was not a common cause of acute biliary infections, and was found in less than 1% of blood and bile samples of patients with acute cholangitis, in agreement with a previous study [18].

In our cohort we found a high prevalence of *E. faecium* and *P. aeruginosa* rising the doubt to change our first line antibiotic prophylaxis. However, we do not think that these species were selected by one dose antibiotic prophylaxis before the procedure. In addition, we did not find any significant correlation between type of antibiotic used as first line prophylaxis and risk of cholangitis and sepsis. On the other hand, a broad spectrum antibiotic prophylaxis would not be advisable because it could select multidrug antibiotic resistant germs. First line antibiotic prophylaxis should take into account the past microbiological history of transplanted children and ad hoc prophylaxis should be done only in selected cases.

In case of cholangitis, the choice of antibiotic therapy should always be guided by culture results (blood cultures, aspiration, or drainage fluid), with the attempt to avoid broad-spectrum antibiotics whenever possible. The choice should take into account biliary drug concentrations, side effects, and potential interactions with concomitant immunosuppressive therapies [11]. In our study, optimization therapy was needed in 20% of cases, and it was based on positive microbiological tests in more than half of cases.

Furthermore, several studies recently described an alarming rise of infections with multidrug-resistant (MDR) pathogens in liver transplant recipients [19,20]. These include extended-spectrum beta-lactamase (ESBL) and carbapenemase producing Gram-negative bacteria (*E. coli*, *K. pneumoniae*, *P. aeruginosa,* and *Acinetobacter baumannii*), as well as methicillin-resistant *Staphylococcus aureus* (MRSA) and vancomycin-resistant *Enterococcus* sp. (VRE) [21,22].

In our experience, among MDR germs, carbapenemase producing Gram-negative bacteria, ESBL and VRE were identified in 22% of our study population.

Optimal duration and route of antimicrobial therapy for patients with acute cholangitis is still controversial. Uno et al. compared retrospectively the outcomes of patients with bacteremic acute cholangitis due to Gram-negative bacilli who received antimicrobial therapy for either 14 or 10 days [23]. There were no differences between the two groups in 30-day mortality and recurrence rate within 3 months. In our study, the mean antibiotic therapy due to post-procedural complications was 9.6 days for observation and supportive treatment. No fatal events were reported.

In conclusion, PTC was associated with a relatively high rate of post-procedure cholangitis, although low rate of sepsis and no fatal events were reported in liver transplanted children. Blood cultures identified the pathogens in roughly a quarter of the cases and were helpful in optimizing the antibiotic treatment. A stepwise use of antibiotics allowed to control the infections without a significant emergence of multiresistant microorganisms.

We presented the first pediatric study focusing on the incidence of cholangitis and sepsis in a large cohort of liver transplant recipients after percutaneous cholangiography (PTC). Blood culture was performed in all post-procedural complications. Post procedure antibiotic therapy was guided by positive blood isolations in more than half a percent of the cases.

Nevertheless, our study had some limitations. It was a single center study that included children with different ages and underlying diseases. Furthermore, due to the retrospective nature of the study, bile cultures were available in only 42% of complicated procedures. Larger prospective trials are needed to identify microbiological growth and optimize empiric or targeted antibiotic treatment in biliary tract infections after PTC in transplanted children.

## 4. Materials and Methods

### 4.1. Study Design and Clinical Definitions

This study was a single-center retrospective study. The inclusion criteria were all transplanted children (patient age <18 years) who performed a PTC from January 2017 to October 2020 in case of suspected biliary strictures. Children that underwent additional non-biliary procedures in addition to PTC, such as percutaneous liver biopsies, were excluded. The primary outcome of our study was to determine the incidence of PTC-related infectious complications in transplanted children, while the secondary outcome was to identify their precise aetiology.

The electronic medical records of all patients were reviewed. Data collected included the demographic and clinical features of study population: age at PTC, gender, underlying disease, type of liver transplant and biliary anastomosis, PTC procedure (including bilioplasty), immunosuppression therapy (including steroids), presence of central venous catheter (CVC), type of antibiotic prophylaxis, microbiological isolation (blood and/or bile culture) in case of cholangitis with relative antibiotic strain susceptibility.

PTC was performed in case of elevated alanine aminotransferase (ALT) or gamma glutamyl transpeptidase (GGT) and/or history of cholangitis and/or liver biopsy consistent with cholangiolar proliferation. Remarkably, in a few cases, PTC was done even in absence of biochemical abnormalities based on evidence of isolated and significant cholangiolar proliferation at liver biopsy [2]. One hour before the procedure, all patients received iv antibiotics classified into first- and second-line prophylaxis. The first one included Cefotaxime, Piperacillin-Tazobactam, and Ciprofloxacin. Second-line antibiotic prophylaxis, such as Meropenem, Daptomycin, and Vancomycin was instead used in cases with previously positive microbiological isolation.

In addition, if they were taking oral prednisone as immunosuppression therapy, they received hydrocortisone for adrenal insufficiency prophylaxis before the procedure.

Cholangitis was defined as fever (>38.5 °C) and elevated inflammatory markers after PTC, while sepsis included hemodynamic instability in addition to cholangitis.

Optimization antibiotic treatment was the switch from first to second line antibiotic treatment or other antibiotic changes based on clinical features or positive microbiological tests.

### 4.2. Bacterial Isolation, Identification, and Drug Susceptibility Test

Blood cultures for aerobic and anaerobic germs were collected from all patients with cholangitis and sepsis, while bile cultures were sent at the physician’s discretion.

Blood and bile samples (bioMerieux-France) were incubated for 5 days at 35 °C, except selected cases where the incubation process was prolonged until 14 or 28 days based on clinical suspect of slow growth germs.

Matrix-Assisted Laser Desorption Ionization-Time of Flight mass spectrometry (MALDI-ToF) using VITEK® MS bioMerieux sa, Marcy l’Etoile, France was applied to identify bacteria and yeasts. Antimicrobial Susceptibility Testing (AST) results were generated using VITEK2® analyzer and E test method (bioMerieux). In selected cases, broth microdilution system SensitreTM DKMGN (Thermo Fisher Scientific, Loughborough, UK) and agar dilution methods for Fosfomycin susceptibility testing AD (Fosfomycin 0.25–256, Liofilchem srl, Roseto degli Abruzzi, Italia) were applied. AST interpretation was done according to European Committee on Antimicrobial Susceptibility Testing (EUCAST) clinical breakpoints and methodology (The European Committee on Antimicrobial Susceptibility Testing. Breakpoints tables of Interpretation of MICs and zone diameters, Version 9.0, 2019 and Version 10.0, 2020).

MDR pathogens were defined as acquired non-susceptibility to at least one agent in three or more antimicrobial categories. They included carbapenemase producing Gram-negative bacteria, vancomycin-resistant Enterococcus sp. (VRE) found in rectal swab and extended-spectrum beta-lactamase (ESBL) isolated either in blood or bile culture.

Furthermore, Carbapenemase producing Gram-negative bacteria were isolated phenotypically and recently genetically too.

### 4.3. Percutaneous Transhepatic Cholangiography (PTC)

PTC was carried out in the interventional radiology suite with the patient maintained under general anesthesia. It was classified as *de novo* in case of first liver puncture or *exchange* if performed through a pre-existing biliary drainage. If PTC showed a biliary stricture, it was dilated with a standard angioplasty balloon (4–8 mm in diameter and 20–40 mm in length) and an internal-external biliary drainage catheter was inserted and maintained in place for 4–6 weeks, see Figure 1. An exchange PTC was performed after 4–6 weeks to further dilate the duct in case of persistent stricture at cholangiogram, and the catheter was then removed. However, in case of persistent stricture, more than one exchange PTC was performed in the same patient until the stricture was solved.

### 4.4. Statistical Analysis

Descriptive statistics are presented at the “procedure” level; categorical variables are summarized with absolute and relative frequencies. P-values from univariate comparisons (chi-square/Fisher test for categorical and t-test sign-rank test for quantitative) are reported. Analysis of predictors on categorical binary outcomes and estimates of incidence were performed by mixed effect logistic regression with patient as random effect to account for the repeated measures design. The entire analysis was performed on Stata 16.

## Figures and Tables

**Figure 1 antibiotics-10-00282-f001:**
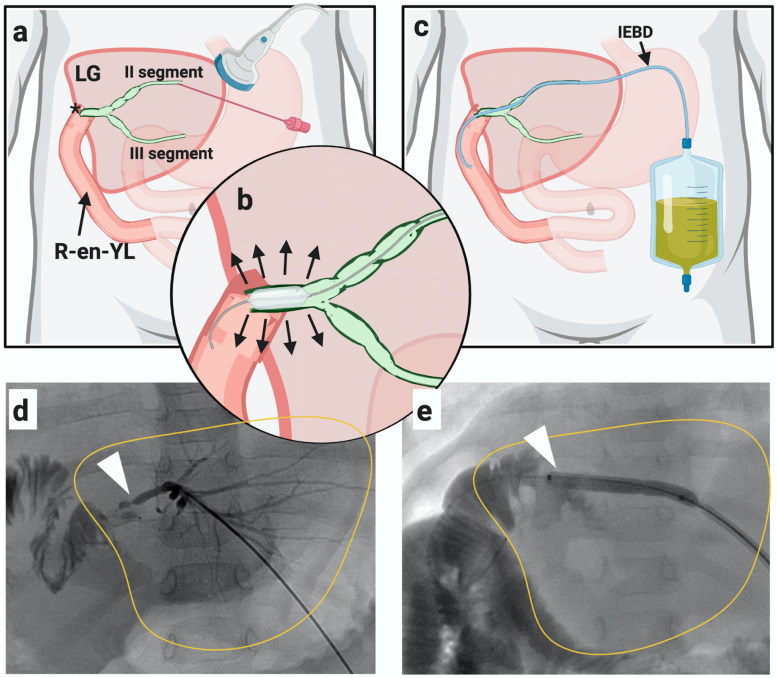
Percutaneous transhepatic cholangiography (PTC). (**a**–**d**): Liver puncture was done through sonographic guidance (PTC de novo) and biliary stricture was identified. (**b**–**e**): Biliary stricture dilatated with a balloon (bilioplasty). (**c**): Internal-external biliary drainage (IEBD) placement. Through the same catheter, a new PTC (exchange) was later done.

**Table 1 antibiotics-10-00282-t001:** Demographic and clinical features of study population.

Age at PTC, Median (Range IQR), Years	3.05 (3–9)
Male gender, N (%)	25 (50%)
Underlying disease	
Biliary atresia, N (%)	33 (66%)
Genetic cholestasis, N (%)	7 (14%)
Metabolic diseases, N (%)	1 (2%)
Miscellanea, N (%)	9 (18%)
Type of LT	
Left lateral segment	46 (92%)
Right lateral segment	2 (4%)
Whole liver	2 (4%)
Type of biliary anastomosis	
Roux-en-Y hepaticojejunostomy	48 (96%)
Duct to duct	2 (4%)
CVC (N%)	17 (34%)
Additional immunosuppressive therapy (steroids use)	25 (50%)
PTC	
De novo	74 (47%)
Exchange	83 (53%)
Bilioplasty (N%)	112 (71%)
Follow up time, median (range IQR), years	2.09 (1–4)

CVC: Central Venous Catheter, LT: Liver Transplant; PTC: Percutaneous Transhepatic Cholangiography, IQR: interquartile range.

**Table 2 antibiotics-10-00282-t002:** Multivariate analysis for risk factors of cholangitis.

Risk Factor	*p*-Value	95%CI	OR
Age at PTC	0.213	0.87–1.03	0.95
Bilioplasty	0.003 *	1.55–8.27	3.58
Steroids	0.868	0.56–1.98	1.05
CVC	0.055	0.13–1.02	0.36
Previous microbiological isolation	0.388	0.64–3.14	1.42
De novo PTC	0.403	0.36–1.51	0.73

CVC: Central Venous Catheter, PTC: Percutaneous Transhepatic Cholangiography, CI: Confidence Interval, OR: Odds Ratio. * *p* < 0.05.

**Table 3 antibiotics-10-00282-t003:** Germs isolated in at least one microbiological test.

Microorganism	Number of Times	Positive Blood Culture (n%)	Positive Bile Culture (n%)
*Enterococcus faecium*	15	4/15 (27%)	11/20 (55%)
*Pseudomonas aeruginosa*	10	4/15 (27%)	6/20 (30%)
*Escherichia coli*	5	2/15 (13%)	3/20 (15%)
*Klebsiella pneumoniae*	4	3/15 (20%)	1/20 (5%)
*Candida albicans*	4	0/15 (0%)	4/20 (20%)
*Staphylococcus aureus*	2	0/15 (0%)	2/20 (10%)
*Enterobacter aerogenes*	2	1/15 (7%)	1/20 (5%)
*Enterococcus faecalis*	2	1/15 (7%)	1/20 (5%)
*Enterobacter cloacae*	1	1/15 (0%)	1/20 (5%)
*Acinetobacter baumanii*	1	1/15 (7%)	0/20 (0%)

## Data Availability

The data presented in this study are available on request from the corresponding author.

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
