# Peer review of "Incidence of Cholangitis and Sepsis Associated with Percutaneous Transhepatic Cholangiography in Pediatric Liver Transplant Recipients"

_antibiotics, 2021, doi:10.3390/antibiotics10030282_

Round 1

Reviewer 1 Report

The manuscript by Sansotta et al, describes the incidence of cholangitis and sepsis associated with PTC in pediatric liver transplant recipients, and the identification of the microbial infection.

The manuscript is of great relevance for the field of pediatric liver transplantation. It is written in a clear and concise style. Conclusions made are appropriate.

Comments

  1. The authors list in lines 51-53 the first line antibiotics used in the prophylaxis. Which antibiotics were applied most frequently? Please provide data on the percentage of administration of the different antibiotics in the 135 procedures as stated in line 76.
  2. Is there a relationship to be detected between the administration of a specific first line antibiotic and the frequency of microbial species detected. Might there be an optimal first line antibiotic administration regimen for prevention?
  3. Table 2: present the Candida positive bile culture as ‘4/20’
  4. One positive blood culture contained two bacterial species; can the authors provide the two bacteria for this culture?
  5. I assume that for the microorganisms grown in the blood and bile cultures antibiograms were made. It would be very valuable to provide the antibiotic susceptibility data for the isolated strains. Please provide these data in a new Table or in Supplementary data.
  6. The authors state that MDR pathogens were isolated. Provide detailed information on these cases. How was a MDR strain defined for this study.
  7. Specify the MDR pathogens (lines 103-105) in more detail, were ESBL producers among them? Highlight these strains in the Table as required under comment 5. Were carbapenemase genes identified (phenotypically and/or genetically); please provide these data.
  8. What antibiotic regimen was applied for the carbapenemase producing infectious agents identified in this study?
  9. Please provide information in Material and Methods on bacterial isolation, identification and drug susceptibility testing.

Author Response

The manuscript by Sansotta et al, describes the incidence of cholangitis and sepsis associated with PTC in pediatric liver transplant recipients, and the identification of the microbial infection.

The manuscript is of great relevance for the field of pediatric liver transplantation. It is written in a clear and concise style. Conclusions made are appropriate.

Thank you very much for your kind feedback

Comments

  1. The authors list in lines 51-53 the first line antibiotics used in the prophylaxis. Which antibiotics were applied most frequently? Please provide data on the percentage of administration of the different antibiotics in the 135 procedures as stated in line 76.

Thank you for your comment. We carefully described the percentage of all antibiotics used as first line prophylaxis, please see lines 78-80

  1. Is there a relationship to be detected between the administration of a specific first line antibiotic and the frequency of microbial species detected. Might there be an optimal first line antibiotic administration regimen for prevention?

Thank you very much for bringing up this important point. In our cohort we found an high prevalence of E. faecium and P. aeruginosa rising the doubt to improve our first line antibiotic prophylaxis. However, we do not think that these species were selected by one dose antibiotic prophylaxis before the procedure. In addition, we do not find any significant correlation between type of antibiotic used as first line prophylaxis and risk of cholangitis and sepsis (p=0.166). From the other hand, a broad spectrum antibiotic prophylaxis would be not advisable because it could select multidrug antibiotic resistant germs. First line antibiotic prophylaxis should take into account the past microbiological history of transplanted children and ad hoc prophylaxis should be done only in selected cases, please see lines 188-195

  1. Table 2: present the Candida positive bile culture as ‘4/20’

Thank you very much. We have added the missing number, please see Table 2

  1. One positive blood culture contained two bacterial species; can the authors provide the two bacteria for this culture?

We have described the blood culture containing two bacterial species. Furthermore, we described in more details bile cultures containing multiple bacterial species as well.Please see lines 112-114, 118-119

  1. I assume that for the microorganisms grown in the blood and bile cultures antibiograms were made. It would be very valuable to provide the antibiotic susceptibility data for the isolated strains. Please provide these data in a new Table or in Supplementary data.

Thank you very much for your question. We have added Supplementary Table 1 describing antibiotic strain susceptibility for the isolated strains either in blood either bile cultures.

  1. The authors state that MDR pathogens were isolated. Provide detailed information on these cases. How was a MDR strain defined for this study.

MDR pathogens were defined as acquired non-susceptibility to at least one agent in three or more antimicrobial categories. They included carbapenemase producing Gram-negative bacteria, vancomycin-resistant Enterococcus sp. (VRE) and extended-spectrum beta-lactamase (ESBL), please see lines 277-280

  1. Specify the MDR pathogens (lines 103-105) in more detail, were ESBL producers among them? Highlight these strains in the Table as required under comment 5. Were carbapenemase genes identified (phenotypically and/or genetically); please provide these data.

Thank you very much for your comment, carbapenemase pathogens were usually identified phenotypically. In the last four years, they were identified genetically too. Please see lines 281-282 We have added in Supplementary table 1 all MDR pathogens.

  1. What antibiotic regimen was applied for the carbapenemase producing infectious agents identified in this study?

Thank you very much for your feedback. Carbapenemase producing infectious germs were treated with a combination of high dose Meropenem and Tigecycline. We also commented on antibiotic therapy administered in case of VRE, please see lines 141-143

  1. Please provide information in Material and Methods on bacterial isolation, identification and drug susceptibility testing

Bacterial isolation, identification and drug susceptibility test were added in Material and Methods paragraph, according to your suggestion. Please see lines 264-276.

Reviewer 2 Report

The authors investigated the incidence of PTC-related infectious complications and identify the microbiological etiology in transplanted children. They concluded that PTC is associated with a relatively high rate of post-procedural cholangitis, although with low rate of sepsis and no fatal events. Blood cultures allowed to find a precise etiology in roughly a quarter of the cases, showing prevalence of Enterococcus faecium and Pseudomonas aeruginosa.

My concerns are as follows:

  1. Above all the presentation of the study is unusual. After the introduction, the authors presented the results and then the discussion. Methodology is presented after discussion. The methodology should be presented before results. Is there a reason for such a presentation of the study?
  2. Limitations of the study should be updated. This was a single-center study, children with different ages and diagnosis were included… All of that should be mentioned in study limitations
  3. Methodology – Primary and secondary outcomes of the study should be clearly stated in methodology of the manuscript.
  4. Methodology – Study protocol should be described in more details. Which parameters (including baseline patient’s characteristics) were recorded? A Table should be provided.
  5. Methodology – Inclusion and exclusion criteria should be clearly stated.
  6. Methodology – Whether the study was approved by Ethics Committee. Please provide IRB number in methodology.
  7. Table 1 – Each abbreviation used in Table should be listed in the legend of the Table.

Round 2

Reviewer 1 Report

The authors have appropriately addressed all the issues I raised in my report.

There are some small text editing issues:

  • first letter of antibiotics is not written with a capital, unless at the start of the sentence.
  • Please check that all latin names of microorganims are in italics
  • Replace the word "germ" by "microorganism"
  • Replace "Antibiotic strain susceptibility was decribed" by "Antibiotic susceptibility testing results of the bacterial isolates are listed in"

Reviewer 2 Report

The authors performed all requested corrections.

Manuscript can be accepted in present form.